# Effects of Land-Use Change on the Community Structure of the Dung Beetle (Scarabaeinae) in an Altered Ecosystem in Southern Ecuador

**DOI:** 10.3390/insects12040306

**Published:** 2021-03-30

**Authors:** Vinicio Carrión-Paladines, Andreas Fries, Andrés Muñoz, Eddy Castillo, Roberto García-Ruiz, Diego Marín-Armijos

**Affiliations:** 1Departamento de Ciencias Biológicas, Universidad Técnica Particular de Loja, San Cayetano Alto s/n, Loja 11-01-608, Ecuador; aomunoz@utpl.edu.ec (A.M.); dsmarin@utpl.edu.ec (D.M.-A.); 2Departamento de Geología, Minas e Ingeniería Civil (DGMIC), Universidad Técnica Particular de Loja, San Cayetano Alto s/n, Loja 11-01-608, Ecuador; aefries@utpl.edu.ec; 3Finca de Permacultura, Finca Fina, Solanda, Vilcabamba 110161, Ecuador; eddycastillojimenez@gmail.com; 4Unidad de Ecología, Departamento de Biología Animal, Vegetal y Ecología, Facultad de Ciencias Experimentales, 23071 Jaén, Spain; rgarcia@ujaen.es

**Keywords:** seasonally dry forest (SDF), land-use change, *Scarabaeinae*, dung removal

## Abstract

**Simple Summary:**

This study analyzed the abundance and diversity of dung beetle communities at several disturbed sites in a tropical dry forest ecosystem in southern Ecuador. Dung beetle community diversity indices with different land uses were related to environmental variables (altitude, temperature), soil physicochemical properties, and food supply (manure). The results indicated that the species *Canthon balteatus*, *Dichotomius problematicus*, and *Onthophagus confusus* are abundant in disturbed sites, where soils are generally more compact and less fertile but contain a greater food supply. These findings can help decision makers to identify disturbed areas and to implement adequate policies for sustainable environmental management.

**Abstract:**

This study evaluated the effects of land-use change (L-UCH) on dung beetle community structure (Scarabaeinae) in a disturbed dry ecosystem in southern Ecuador. Five different L-UCH classes were analyzed by capturing the dung beetle species at each site using 120 pitfall traps in total. To determine dung beetle abundance and diversity at each L-UCH, a general linear model (GLM) and a redundancy analysis (RDA) were applied, which correlated environmental and edaphic conditions to the community structure. Furthermore, changes in dung-producing vertebrate fauna were examined, which varied significantly between the different L-UCH classes due to the specific anthropogenic use or level of ecosystem disturbance. The results indicated that soil organic matter, pH, potassium, and phosphorus (RDA: component 1), as well as temperature and altitude (RDA: component 2) significantly affect the abundance of beetles (GLM: *p* value < 0.001), besides the food availability (dung). The highest abundance and diversity (Simpson’s index > 0.4, Shannon-Wiener index > 1.10) was found in highly disturbed sites, where soils were generally more compacted, but with a greater food supply due to the introduced farm animals. At highly disturbed sites, the species *Canthon balteatus*, *Dichotomius problematicus*, and *Onthphagus confuses* were found specifically, which makes them useful as bio-indicators for disturbed dry forest ecosystems in southern Ecuador.

## 1. Introduction

Loss of tropical forests due to land-use changes (L-UCH) is a major environmental concern at local, regional, and national scales, especially in developing countries [1]. Approximately 32% of total tropical forest loss is due to anthropogenic activities [2,3], caused by high population growth, which subsequently leads to landscape fragmentation and a reduction of species diversity due to deforestation or selective logging [4,5]. The main cause of L-UCH is the economic activity of the population, such as timber extraction, or the conversion of forests into agricultural land, together with the ensuring infrastructure construction [6]. Once the natural vegetation cover is converted, exotic species (plants and animals) are introduced into the disturbed areas, which generally do not have native predators [7], and, therefore, can provoke additional changes in the ecosystems [8,9]. This exemplifies that human impacts on natural ecosystems can sometimes be irreversible; however, alterations in local environmental and climate conditions are generally observed [10,11]. L-UCH leads to greater soil erosion and surface runoff, which reduces aquifer recharge and water availability [12,13,14]. Furthermore, soil texture and bulk density change [15], which can reduce soil organic matter content [16] and the availability of macronutrients, such as nitrogen and phosphorus [17].

The estimated environmental and socio-economic trends for tropical forests indicate that large areas of the remaining natural ecosystems are endangered by L-UCH [18]. However, there are only a few studies which have focused on the resilience of biodiversity within disturbed landscapes [19], and where habitat fragmentation has affected biodiversity for decades [20]. Not only plant species are affected, but also insects and soil organisms, which are placed in danger of extinction, due to the excessive use of polluting chemicals (agriculture), which also eliminate desired species. Therefore, the expansion of agriculture and livestock farming alters the natural conditions of insects and other soil organisms within their habitats, and by which bees (*Apis mellifera*), ants (*Formicidae*), and beetles (*Scarabaeus* sp.) are especially affected [21,22].

A group of insects which is widely used to analyze the effects of habitat modifications upon biodiversity are dung beetles (*Scarabaeinae*) [23], because they are sensitive bio-indicators respective to ecosystem changes and can be easily sampled. Furthermore, dung beetles are widely distributed, and their taxonomy and ecology are relatively well known [24]. In South America, several studies have been conducted on the diversity of dung beetles in different ecosystems. These investigations mainly focused on morphology, feeding habits, and ecological function [24,25,26], besides species composition, richness, and abundance of dung beetles, while taking into consideration altitudinal and successional gradients, as well as seasonal variations (climate) [27,28]. Moreover, these studies illustrated that dung beetles can be used as biodiversity indicators for both undisturbed and disturbed areas, and thus are potential identifiers of highly disturbed ecosystems [29,30].

In general, dung beetles biologically degrade organic materials for their food source, such as mammalian feces, dead animal carcasses, rotten plant matter, etc. [31]. Therefore, they play an important ecological role within the nutrient cycle, influencing seed dispersal and forest regeneration, as well as the control of parasite populations and diseases [32,33]. As Badenshort et al. [34] clarified, dung beetles improve the ecological functions and soil fertility of ecosystems, because they introduce essential nutrients into the root zone, which would otherwise remain on top of the soil, and therefore inaccessible to plants. Furthermore, dung beetles create tunnels, which improve infiltration rates and soil aeration [23]. According to Halffter and Mathews [35], dung beetles (*Sarabaeinae*) are classified into three functional groups: (i) *Telecoprids* or Rollers (food balls are rolled some distance before burial), *Paracoprids* or Tunnellers (tunnels are excavated for food source), and (ii) *Endocoprids* or Inhabitants (feed and reproduce within the food source). Additionally, dung beetles can be classified into nocturnal or diurnal species (e.g., [36]).

However, the effects of L-UCH upon dung beetle communities, as important soil organisms and in semi-arid regions, have rarely been studied [17], especially in tropical dry forests (TDF), where an increase in deforestation rates has been observed during the last few decades [37]. In Ecuador, for example, the deforestation rate is extremely high [38,39], and the Amazon forest and tropical mountain forests are especially affected, along with the coastal and dry forest ecosystems, due to agricultural or livestock farming influences (e.g., [40,41]). The intensification of conventional agricultural practices in these areas is the main reason for the loss in the soil’s carbon sequestration ability and the impoverishment of the soil’s nutrient availability, which, consequently, reduces biodiversity over time [42]. Furthermore, soil degradation leads to lower plant productivity and to higher production costs [43], which is why agricultural frontiers are expanding and a rural exodus is taking place to fulfil the socio-economic requirements of the population [44,45].

In the semi-arid regions of Ecuador, the effect of land-use change (L-UCH) on dung beetle communities has so far not been studied. Only studies with respect to dung beetle abundance and diversity in natural humid and Andean forests have been executed [46,47], besides investigations with respect to the effect of road construction and altitudinal gradients on dung beetle communities [48,49]. The main reason for this is that to analyze the effects of L-UCH on the abundance and diversity of dung beetles requires the implementation of a large number of field plots [50], besides the analysis of alterations in soil properties and environmental conditions (climate). Furthermore, the altered food provision for dung beetles (manure) in different L-UCH classes must be considered.

Therefore, the objective of the present research was to analyze the disruptive effects on the abundance and diversity of dung beetle communities in areas with different levels of disturbance (L-UCH). Therefore, edaphological (soil) and environmental factors (climate and altitude) were analyzed, applying a general linear model (GLM) and a redundancy analysis (RDA). In addition, the relative manure supply of native and introduced vertebrates was examined to contrast the food availability for dung beetle communities among the different L-UCH classes. The results might help decision makers to identify highly degraded ecosystems, where adequate policies and sustainable environmental management should be implemented [51].

## 2. Materials and Methods

### 2.1. Study Area

The investigation was carried out in the parish of Malacatos (Loja, Ecuador), located between 4°12′24″ and 4° 13′39″ S (south), and between 79°19′12″ and 79°19′21″ W (west) (Figure 1). The altitude varies from 1366 m to 1968 m a.s.l, and topography is characterized by moderate to strong slopes [52]. The climate in the study area is semi-arid, with an average annual temperature of 20 °C and an average annual precipitation of about 800 mm [53], and where mean annual temperature decreases with altitude, whereas annual precipitation increases towards higher elevations. The precipitation shows a clear annual cycle in the study area, with one main rainy season from October to April (austral summer), when 88% of the total annual rainfall amounts are measured, and a dry season from May to September (austral winter; [43]).

The natural vegetation in the study area is tropical dry forest (TDF) [40]. However, as is the case for most forest ecosystems in Ecuador, large areas of TDF are disturbed by anthropogenic activities, mainly related to logging for migratory agriculture or selective logging. As a result, TDFs contain a wide variety of disturbances, which can be classified into the following main L-UCH classes: slightly disturbed TDF, pasture or grassland, and agriculture, in which the dominate crop is coffee, as well as other monoculture systems and agroforestry [40,54].

The specific L-UCH classes chosen for this study are specified subsequently (see also Figure 1):(a)Coffee in an old agroforestry system (CoAS), where cultivation of coffee (*Coffea arabica*) is associated with banana (*Musa paradisiaca*), papaya (*Carica papaya*), and guavas (*Inga feuilleei*), but also with native trees species, especially faique (*Acacia macracantha*), wilco (*Anadenanthera colubrina*), and fig trees (*Ficus* sp.). In this L-UCH class, native vertebrates exist, such as deer (herbivores; *Odocoileus peruvianus*), squirrels (omnivores; *Sciurus* sp), and anteaters (insectivores; *Tamandua* sp), but also domestic animals, particularly goats (herbivore; *Capra hircus*).(b)Seasonally disturbed dry forest (SDF), in which the dominant tree species are wilco (*Anadenanthera colubrina*, with an average DBH of 0.26 m and an average height of 18 m), ceibo (*Ceiba trichistandra*, with an average DBH of 1 m and an average height of 20 m), and faique (*Acacia macracantha*, with an average DBH of 0.21 m and an average height of 17 m). In this L-UCH class native vertebrates dominate, especially anteaters, deer, skunks (non-strict carnivores and omnivores; *Conepatus semistriatus*), and opossums (omnivores; *Didelphis marsupialis*), while domestic animals are also frequent, such as goats, cows (obligate herbivore; *Bos taurus*), and horses (non-ruminant herbivore; *Equus caballus*).(c)Organic crop area (OCA), which is a polyculture system, including chard (*Beta vulgaris*), turmeric (*Curcuma longa*), cabbage (*Brassica oleracea*), and carrot (*Daucus carota*). In this L-UCH class domestic animals are more frequent, such as donkeys (herbivore; *Equus asinus*) and horses, but some native species, particularly deer and skunk, can also be found.(d)Grasslands (GLd), mainly composed of yaragua grass (*Hyparrhenia rufa*) and *Cybitax antisyphilitica* seedlings. In this L-UCH class generally only domesticated animals exist, such as cows, horses, and donkeys.(e)Coffee in a young agroforestry system (CyAS), which were recently planted (max. 3 years ago), including coffee (*Coffea arabica*), apple (*Malus domestica*), and guava (*Inga feuilleei*), but also young trees, especially arabisco (*Jacaranda copaia*), faique (*Acacia macracantha*), and wilco (*Anadenanthera colubrina*). In this L-UCH class native vertebrates, such as anteaters, skunk, and deer can be found, but the presence of domestic animals is higher, in particular cows and horses.

### 2.2. Soil Sampling and Analytical Methods

Sampling was performed between May 2019 and February 2020. Mean altitude for each L-UCH class was determined using a GPS (Garmin eTrex Legend). Additionally, air temperature (°C) at the individual L-UCH classes was measured during the whole study period by means of a HOBO sensor (UA-001-08, HOBO Pendant^®^, Bourne, MA, USA).

The identification and boundary of each L-UCH class was determined by field excursions and visualized using the Google Earth Pro program [55] (Figure 1). The plots in each L-UCH class were implemented according to the accessibility and considering a distance of 150 m to the L-UCH boundary, to guarantee homogeneity of the sampling. In each L-UCH class, three plots of 20 m × 20 m were installed, which were evenly separated with a distance of 20 meters between each one. Soil samples were taken at the limits and in the center of each plot at a depth of 10 cm using standardized metal cores (diameter 7 cm; [56]). For the analysis of the soil properties, 2 samples were extracted at each sampling point (one for determination of the bulk density (Bd), and another for chemical analysis). This resulted in 10 individual samples per plot (5 Bd and 5 chemical analysis) and 30 soil samples for each L-UCH class. After sample extraction, they were packed in separate plastic bags, labeled, and transported to the soil laboratory of the Universidad Técnica Particular de Loja (UTPL), where the samples for the chemical analysis were stored at 4 °C in a refrigerator for later analysis [57], whereas the samples for the Bd analysis were air dried at room temperature for 24 h.

In the laboratory, first, Bd was determined by means of the cylinder method [58], for which the individual Bd samples were oven-dried for 48 h at 105 °C. Subsequently, the cooled samples destined for the determination of soil texture and soil chemical properties were dried at room temperature, all visible root debris removed, and then the samples were sieved through a 2-mm mesh [43]. Soil texture of the individual samples was determined by means of the hydrometer method [59], whereas soil pH was measured by a pH-meter applying the standard method [58]. Furthermore, soil organic carbon (SOC) was determined applying the Walkley and Black method [60], for which a thermostat at 125 °C was inserted for 45 min after the samples were oxidized with a K_2_Cr_2_O_7_/H_2_SO_4_ solution. Soil organic matter (SOM) was estimated based on the Tyurin method [61,62], and total nitrogen (TN) was determined using the Kjeldahl method [63]. Finally, the phosphorus content and the potassium content were determined using the atomic absorption spectrophotometry method [64].

All individual results were averaged to obtain mean values for each L-UCH class (Bd, soil texture, and chemical properties), and the standard deviation for each L-UCH was also calculated.

### 2.3. Sampling, Identification, and Categorization of Dung Beetles

To analyze the composition of the dung beetle communities for the different L-UCH classes (CoAS, SDF, OCA, GLd, and CyAS), two transects were established within each plot, separated by 7.5 m. Then, 4 pitfall traps were installed along each transect, separated evenly by 4 m, which resulted in 8 pitfall traps per plot and a total of 120 pitfall traps for all L-UCH classes [65]. The pitfall traps were buried, ensuring that the top of each trap was equal to the soil surface, and then, they were partially filled with a mixture of water and neutral soap to preserve the insects. Finally, a plastic spoon was placed on top of each trap, which contained two types of bait. One bait consisted of 20 g of pig manure (4 pitfall traps per plot), while the other bait included a carrion mixture and pieces of fish skin (20 g; 4 pitfall traps per plot). The pitfall traps were baited only once.

The sampling was carried out in December 2019 during the rainy season, in which the pitfall traps remained active for 48 h, as recommended by da Silva and Hernández [65]. The trapped insects of each pitfall were collected manually and put inside of Ziplock bags, filled with 70% ethyl alcohol to conserve the dung beetles [49,66]. All bags were labeled and transported to the Insect Laboratory of the UTPL (Colección de Insectos Sur del Ecuador [CISEC-MUTPL]) for further analysis. In the CISEC-MUTPL all captured dung beetles were classified taxonomically, using dichotomous keys, and afterwards quantified considering the different L-UCH classes [30]. After the identification, classification, and quantification, photographic records were taken and the best specimens of dung beetles were stored in entomological boxes.

### 2.4. Manure-Producing Vertebrates in the Different L-UCH Classes

To obtain information with respect to the type of manure and its amount within the different L-UCH classes, questioning of the land owners (farmers) was executed [67], besides direct observations during the field campaigns. The questioning was based on number and type of animals (domestic or wild) that inhabit or circulate in the different L-UCH classes. However, the quantity of native vertebrates that inhabit the different L-UCH classes was not known by the farmers, and for the number of domestic animals circulating only mean values were specified. Therefore, only the type of manure was included to analyze abundance and diversity of dung beetle communities at the different L-UCH classes. This information was summarized applying descriptive statistics, for which a scoring scale was established to identify the type of animals/manure present at each L-UCH class. The scale ranges between 1 and 0, which indicates the percentage of confirmation by the land owners that these animals/manures exist within the specific L-UCH class.

### 2.5. Data Analysis

The data from physical–chemical soil analysis of the different L-UCH classes were subjected to a one-way analysis of variance (ANOVA, F test, *p* < 0.05). If the statistical analysis was significant, the means were compared to the Tukey’s post hoc test and accepted with a *p*-value of <0.05 in all cases. The dung beetles captured in the pitfall traps were similarly statistically analyzed (ANOVA and Tukey’s post hoc test), considering the total number of each species and the L-UCH class. All analyses were carried out using SPSS statistical software (v.15.0; SPSS Inc., Chicago, IL, USA).

The abundance and diversity of dung beetles were calculated applying the Shannon–Wiener and the Simpson indexes [68,69], using the PAST software package [70]. To determine the differences between the L-UCH classes, the Shannon–Wiener and the Simpson indices were analyzed by means of the ANOVA. To corroborate the species abundance of the different L-UCH classes, the number of species found and the Shannon–Wiener index were used, extrapolating these data through Hill’s numbers [71,72]. By means of the extrapolated data, the diversity pattern of the dung beetle communities could be established, using the “iNEXT” package [73,74,75] of the R 3.1.1 program (R Core Team, 2014) [76].

To determine the effects of abiotic factors (altitude, temperature, and soil physical–chemical properties) upon the dung beetle communities, a generalized linear model (GLM; [27]) was applied, which quantified the relevance of the individual abiotic factors with respect to the number of beetles found at each L-UCH class. Additionally, by means of XLSTAT software [77], a redundancy analysis was performed, which verified the existence of a linear relationship between the abundance of the dung beetles and the abiotic factors.

## 3. Results and Discussion

### 3.1. Abiotic Conditions at the Different L-UCH

The average altitude of each L-UCH class, as well as the measured mean air temperature during the study period (May 2019–February 2020), are summarized in Table 1. As is commonly known, air temperature decreases with altitude in the troposphere; however, in the study area air temperature also depends on L-UCH classes. Strongly disturbed sites (CoAS, OCA, GLd, and CyAS) showed higher temperatures compared to slightly disturbed natural vegetation (SDF) at the same altitude or even at higher elevations (Table 1). This is likely due to the lack of a dense canopy layer, as clarified by Fries et al. [78], who studied the effect of land use changes in a tropical mountain forest in southern Ecuador. They found that the canopy layer shelters the air inside the forest from temperature extremes, which results in generally lower mean temperatures (about 1 °C) inside the forest stands compared to disturbed sites (see Figure 1). During sunny weather conditions (e.g., dry season), the differences are even greater (about 3 °C). However, extreme differences were reported for daily maximum temperatures, which showed deviations up to 15 °C when comparing forest stands with grassland. The same is valid for air humidity, which is notably higher inside forests, when compared to open sites [10].

With respect to annual precipitation, the differences between the L-UCH classes were negligible (average 2006–2015: 1030 mm year^–1^ to 1050 mm year^–1^; [43]), due to the location of the study sites at the same slope, and within an altitudinal range of 500 m. However, due to the orographic precipitation effect [78,79], precipitation increases towards higher elevations [80], which was confirmed by information from the official climate station of the Ecuadorian weather service (Instituto Nacional de Meteorología e Hidrología (INAMHI); station Malacatos) located at the valley bottom, which showed an annual mean precipitation of only 800 mm [53].

During the study period (May 2019 to February 2020) a total precipitation amount of 550 mm was measured at the Malacatos station, whereas during the beetle collection period (December 2019, rainy season) monthly precipitation amounted to approximately 80 mm. This indicated that water stress was unlikely to affect dung beetle abundance nor diversity in this study, as reported by other investigations during dry seasons [49,81]. Therefore, the other environmental factors (soil properties, altitude, and temperature, [49]) and the level of disturbance (L-UCH class) have a greater influence on the development of dung beetle communities, with food availability (manure).

### 3.2. Physical Soil Properties of the Different L-UCH

Soil bulk density (Bd) at a soil depth of 10 cm varied between 0.71 g cm^−3^ and 1.12 g cm^−3^, with significant differences between the L-UCH classes (Table 2). Bd was highest for GLd (1.12 g cm^−3^) and CyAS (1.05 g cm^−3^), whereas for SDF, the lowest values were measured. This indicates that highly disturbed sites have more compact soils, which is likely due to intensive anthropogenic activities, such as the overgrazing of cattle (*Bos taurus*; GLd) or the use of agricultural machinery (CyAS). Soil compaction is especially detectable within the first few centimeters of the soil (e.g., Ayoub et al. [82]), which leads to faster erosion and a decrease in the water retention capacity of the soils, as well as in soil fertility [10]. Therefore, soil compaction can decrease growth rates and crop productivity at affected sites [83,84].

In contrast, lower Bd values were obtained for less disturbed L-UCH classes (OCA: 0.92 g cm^−3^; CoAS: 0.80 g cm^−3^, and SDF: 0.71 g cm^−3^, respectively), which is due to the lower anthropogenic pressure upon these soils [85]. However, as Conti et al. [86] found in conserved soils under primary forests in Argentina, Bd values under natural vegetation (1.22 g cm−^3^) can be higher than those obtained in the study area (Table 2), which could be related to the amount of soil organic matter (SOM) produced by the different vegetation types (Table 3). As shown by Murray et al. [87], in organically managed agroforestry systems, soil compaction is lower due to the annual contributions of leaf litter, exudates, and root biomass, which increase the SOM content in the soil and modify the physical, chemical, and biological properties [88]. Thus, the relatively high SOM content in the study area (SDF: 4.5%; OCA: 3.7%, and CoAS: 6.03%, Table 3) improves the soil structure and increases porosity, particularly within the first 10 cm [87,89].

The dominant soil texture of the L-UCH classes was loam-sandy (Table 2), for which little information is available. Only Ben Dkhil et al. [90] used this soil texture type as a substrate for potato cultivation, and found no benefit in plant yield compared to other texture types. However, for soils with similar textures, such as sandy or sandy-loam soils, an abundance of scientific literature exists. For example, Yost and Hartemink [91] analyzed Bd values for sandy soils, which were similar to the values obtained in this study (Table 1). They determined an average Bd of 1.30 g cm^−3^ for the soil surface layer (horizon A), while the SOM amount was high. This was confirmed by Dam et al. [92], who found a Bd of 1.29 g cm^−3^ in sandy-loam soils at depths between 0–10 cm. However, Bd is also influenced by tillage practices. As Kushwaha et al. [93] indicated by comparing zero tillage and residue applications with specific conventional tillage practices, Bd was lower under reduced-tillage (1.27 g cm^−3^) compared to conventional tillage (1.40 g cm^−3^) at sandy-loam sites. This is consistent with the present study, where differences in Bd were detected where tillage was practiced. Furthermore, a recent study in Ecuador showed, if vermicompost was used as an organic amendment within sandy soils, after 45 days of decomposition not only was Bd improved, but also less CO_2_ was emitted when compared to loamy soils [94]. According to Lozano-García and Parras-Alcántara [95], these differences and improvements in the soil (including Bd) were probably due to high levels of SOM, adequate levels of total carbon (TC), and the availability of organic C, which were significantly higher when compared to clay soils.

### 3.3. Chemical Soil Properties of the Different L-UCH

The chemical soil properties of the different L-UCH classes are presented in Table 3. The results were similar to values reported in other studies carried out in Ecuador and Latin America (e.g., [7,8,30]). However, soil pH at the different L-UCH showed significant statistical differences (HSD Tukey), in which SDF, OCA, and CoAS had a soil pH over 6.0, whereas for GLd and CyAS the soil pH was close to 5.0. The lower pH at highly disturbed sites was in accordance with Potthast et al. [96], who measured a soil pH of approximately 5.2 at pasture sites (here: GLd) in southern Ecuador and at soil depths between 5 to 10 cm.

Taking into consideration optimal soil pH ranges [97], the soil pH found at the different L-UCH classes indicated that generally no problems existed with respect to nutrient availability, since P is known to be available within the pH range of 6.0 to 7.0. This excludes GLd and CyAS, where low contents of P were also measured (Table 3). Other micronutrients, such as B, Cu, Fe, Mn, Ni, and Zn, are available within a pH range of 5.0 to 7.0, which indicates that no limitation of their availability could be assumed in the study area. However, for other necessary macronutrients (N, K, Ca, Mg, and S), these soil pH values are suboptimal, since values between 6.5 to 8.0 are required [98]. Therefore, the only L-UCH class with an optimal soil pH was OCA, because of the continuous application of organic and bio-organic fertilizers at this site [99].

The SOM (6.7%), TN (0.34%), and SOC (3.9%) were significantly higher in CoAS than in the SDF (4.5%, 0.23%, and 2.6%, respectively), OCA (3.7%, 0.18%, and 2.2%, respectively), and GLd (2.3%, 0.11%, and 1.3%, respectively), whereas the CyAS soil contained the lowest amounts (2.1%, 0.11%, and 1.2%, respectively; Table 3). These results are consistent with those reported by Tumwebaze et al. [100], who demonstrated that in the old agroforestry system (CoAS) the regular tree pruning and the renewal of roots over the years resulted in the accumulation of organic matter (SOM), which is related to the amount of total soil nutrients. Furthermore, the diversity of plant and wildlife species in CoAs and SDF increases the SOC in tropical agroforestry systems [101]. This was also confirmed by Ochoa et al. [102], who found relatively high levels of SOM (4%) and TN (0.3%) for CoAS in southern Ecuador.

In general, higher biomass production is measured in undisturbed tropical regions [50,103]. This is illustrated by the higher contents of SOM, TN, and SOC within soils under CoAs and SDF in the study area (Table 3). In disturbed systems a reduction of SOM levels, TN, and SOC is expected since decomposition of organic residues is promoted due to the higher temperatures (see Section 3.1), tillage, harvest, or burning practices, which further decrease the physical protection of soil organic matter [104]. However, when comparing the values of CoAS and SDF, lower values of SOM, TN, and TC were found for SDF, which was likely due to the steeper slope gradients (more than 70%), which implies stronger soil erosion processes and the absence of the application of organic fertilizers [105].

### 3.4. Dung Beetle Species Found at the Different L-UCH Classes

Table 4 shows the average number of dung beetles found in the different L-UCH classes. Comparing the different sites, it is obvious that the highest number of individuals were trapped at CyAS, particularly the species *Canthon balteatus* (24.4 ± 7.9), *Dichotomius problematicus* (9.9 ± 4.4), and *Onthophagus confusus* (5.2 ± 2.3). The second L-UCH class with a large number of dung beetles was OCA, where the species *C. balteatus* (5.5 ± 2.0), *D. problematicus* (5.7 ± 1.9), and *O. confusus* (2.3 ± 1.3) were the most abundant. The other L-UCH classes showed a lower quantity of these beetles, with the exception of the species *Onthophagus curvicornis* (6.1 ± 2.0), which was also found in great numbers at CoAS. This indicated that the dung beetle species *C. balteatus, D. problematicus,* and *O. confusus* are more abundant at highly disturbed sites.

Analyzing the species distribution of the individual L-UCH classes (Table 4), considerable differences were found. For example, the abundance of *C. balteatus* varied significantly between the L-UCH classes (*p*-value < 0.001), where the highest values were found in CyAS (24 ± 7.9), whereas in OCA (5.5 ± 2.0) and GLd (5.5 ± 1.9) the number of individuals notably deceased. The lowest numbers of this species were found in CoAS (1.0 ± 0.4) and SDF (0.3 ± 0.6). The same was valid for the species *D. problematicus*, which showed significant statistical differences between the L-UCH classes (*p*-value <0.001). The most individuals of this species were found in CyAS (9.9 ± 4.4) compared to GLD (6.4 ± 1.8), OCA (5.7 ± 1.9), and SDF (4.5 ± 1.7). The third species with a high numbers of individuals at all L-UCH classes was *O. confuses*, which was more abundant at CyAs (5.2 ± 2.3) and CoAS (4.7 ± 2.8), compared to OCA (2.3 ± 1.3) and GLD (1.1 ± 0.4).

The other species present in the study area were *Onthophagus curvicornis*, *Phanaeus achilles*, *Onoreidium ohausi,* and *Aphodius* sp1, but with mostly insignificant numbers for all the L-UCH classes. These results are consistent with Domínguez et al. [49], who found that *C. balteatus*, *D. problematicus,* and *Onthophagus* sp. are the species with greatest abundance in neotropical dry forest, whereas the species *Ph. achilles* and *O. ohausi* are represented in lower numbers.

Regarding the Shannon–Wiener index (H ‘; Table 4), species diversity was relatively low (< 1.2) in all L-UCH classes [64], but with significant differences between the sites (*p*-value < 0.001). Especially at CyAS, species diversity was very low (H ‘= 0.4 ± 0.4), whereas the highest values were calculated for CoAS and SDF (H ‘= 1.1). In contrast, the Simpson index, which indicates habitat diversity, showed the lowest diversity for SDF (0.8 ± 0.3) and highest diversity for OCA (0.3 ± 0.0). This indicated that dung beetle individuals are rare in the SDF, but more numerous in highly disturbed L-UCH classes. This might be due to the continual extraction of timber trees, converting the natural forest into a secondary forest, where faique (*Acacia macracantha*), in particular, is harvested for charcoal, and wilco (*Anadenanthera colubrina*) for logs used in house construction [106]. These activities generally produce a disproportionate amount of litter (leaves and branches), which affects the reproductive success of the beetles by preventing brood relocation [107]). These results agree with Otavo et al. [108] and Noriega et al. [109], who found that the diversity and abundance of dung beetles increased in more disturbed ecosystems (CyAS and GLd), where particularly generalist species, such as *C. balteatus*, were found. Furthermore, at disturbed sites, generally greater amounts of food are available due to more intensive livestock grazing [26]. This was confirmed by the extrapolation method, which estimated distribution patterns of dung beetle communities in the study area, and where SDF was the L-UCH class where less species diversity and a lower number of individuals should be expected (Figure 2).

### 3.5. Influence of Food Availability (Manure) on the Abundance of Dung Beetle Species in the Different L-UCH Classes

Table 5 shows the type of native and domestic animals which inhabit or circulate in the different L-UCH classes. However, as mentioned before (Section 2.4), total numbers of animals could not be determined, only mean values for domestic animals circulating per hectare were provided by the land owners, which is why the abundance of all dung producers could not be estimated. Nevertheless, this information can be used as rough guidance for the abundance of domestic dung producers in the different L-UCH classes, in which values (X) close to 1 indicate frequent occurrence of these animals, while values around 0 indicate absence or less frequent occurrence of these animals.

In general, as confirmed by the land owners, the dominant native animals in SDF are anteaters, opossums, skunks, and squirrels. At the same time, domestic animals, particularly goats, graze in large numbers, while fewer numbers of cows, horses, and donkeys are present in SDF. In more disturbed areas (e.g., GLd), domestic animals such as cows, horses, donkeys, and, to a lesser extent, goats predominate, while the presence of native animals is generally rare. In succession areas (CoAS and OCA) less domestic animals are present, and goats predominate in CoAS, with donkeys as well as horses in OCA. Cows are also numerous in CyAS, possibly because this area is linked to pastures (GLd).

According to the farmers, the use of fresh dung from domestic animals, as well as human feces, has been implemented in the study area over the last 3 years as an ecological alternative for habitat restoration (Table 5). Most commonly used are fresh cow, goat, and chicken manure, and to a lesser extent human feces, as well as horse or donkey manure. The fresh dung is particularly used in CoAS and CyAS as a fertilizer for tree seedlings, as well as in OCA and GLd as a fertilizer for crops. These types of manure (animal and human) are applied directly to the soil, generally without going through a composting process, and without a pattern or technical scheme of use, which is why it must be considered as food for the dung beetle communities. Sometimes repeated applications in the same area were observed, which leads to overdoses and accumulation of fresh manure, provoking soil and air contamination [110].

As mentioned above, *C. balteatus, D. problematicus,* and *O. confusus* (Figure 3a–c, respectively) were present in all L-UCH classes, but they were more abundant in highly disturbed sites (CyAS and GLd). These species are generalist (eurytopic) [111,112]), and are adapted to consume cattle, horse, goat, and human feces [113,114]. Therefore, the abundance of food resources should be considered as an important constraint for dung beetle distribution, but also the type of manure is relevant, since native species generally do not consume manure from domestic animals [115]. In this context, native species are likely to be more affected by land use changes than generalist species. Likewise, the low population or disappearance of native animals in highly disturbed areas is a factor that affects the abundance of certain native dung beetle species [116], because the availability of manure (food) from domestic animals allows generalist species communities to grow, while concurrently, native species are reduced [117].

This explains why the native Ecuadorian species *Onoreidium ohausi* (Figure 3e) and *Phanaeus achilles* (Figure 3g) [118] were the least abundant in the study area; although little information about these species exists [119], the latter has also been described as eurytopic, with a high tolerance to anthropogenic interventions [120]. However, it is possible that *Ph. achilles* was less abundant because this species is not adapted to inhabit compact soils, but this is a question for future research.

### 3.6. Effects of Abiotic, Edaphic, and Type Factors of L-UCH on the Total Abundance of Scarabaeinae Species

Altitude (Z = 20.584, *p* < 0.001) and temperature (Z = 21.32, *p* < 0.001) were positively related to total dung beetle abundance (Table 5), particularly for the L-UCH classes CoAS (Z = 72.466, *p* < 0.001), CyAS (Z = 16.323, *p* < 0.001), and OCA (Z = 2.488, *p* < 0.001), whereas SDF showed a negative relationship (Z = −7.493, *p* < 0.001), and GLd did not show any relationship (Z = 0.078, *p* < 0.937). Chemical properties such as pH (Z = −15.880, *p* < 0.001), SOM (Z = −14.480, *p* < 0.001), TN (Z = −13.980, *p* < 0.001), TC (Z = −14.430, *p* < 0.001), K (Z = −10.85, *p* < 0.001), P (Z = −10. 48, *p* < 0.001), and C/N ratio (Z = −1.343, < 0.001) had a negative relationship with dung beetle abundance, while Bd (Z = 15.870, *p* < 0.001) showed a positive relationship. The three abiotic factors (altitude, temperature, and L-UCH class) determined the specific environmental conditions at each site (Table 1), which is why generalist species, *such as C. balteatus, D. problematicus,* and *O. confusus* exist in greater abundance at disturbed sites. At higher elevations temperatures decrease, but precipitation increases, leading to slower decomposition of organic matter [72], which also includes carcasses and animal manure [10,78,121], and therefore the availability of food for dung beetles should be greater. However, other studies conducted in Bolivia [122] and Mexico [123] reported a gradual decrease in the total abundance of dung beetles with increasing altitude, which could be due to the more pronounced temperature variation between seasons (summer and winter), but also due to the lower biomass production at higher altitudes, which also includes the biomass of manure producers. On the other hand, annual rainfall and, consequently, soil moisture must be taken into account, as they determine survival rates among dung beetle larvae [124]. However, in the study area, no significant differences were found in the annual precipitation between the different L-UCH classes, and the collection of beetles was carried out during the rainy season, so that a deficit in the availability of water could not explain the decrease in dung beetle population and abundance at higher elevations.

Another factor that was determined to impact the abundance of the dung beetle was Bd (Table 2). The highest number of dung beetles were found in CyAS and GLd, where simultaneously Bd was highest (1.05 and 1.12 g cm^−3^ respectively). In these areas the eurytopic species *C. balteatus, D. problematicus* and *O. confusus* had the greatest abundance, which is consistent with the work of Brussaard and Slager [125], who clarified that Bd affects the reproduction of the dung beetle. In less compact soils (lower Bd) the development of breeding and nesting is limited, because the females need more time for the construction of the egg chamber and the construction of the brood chamber, for which reason dung beetles generally are more abundant in compact soils [65,126]. Furthermore, one of the ecological functions of dung beetles involves the reduction of soil Bd, thus allowing greater porosity, aeration, and infiltration of water into the soil [34,127], as well as aiding in the decomposition of organic matter and the incorporation of dung for nutrient recycling [128,129]. Therefore, the species *C. balteatus, D. problematicus,* and *O. confusus* are found in particularly large numbers at highly disturbed sites (CyAS, and GLd; Table 4). However, more compacted soils may limit the performance of the dung beetles to incorporate organic matter into the soil; but once these physical soil barriers are overcome, dung beetle activity improves the chemical soil properties, providing more nutrients to plants [130,131].

Although there is no direct relationship between the total abundance of dung beetles and the chemical properties of the soil (Table 6), the generalist species (*C. bateatus, D. probematicus,* and *O. confusus*) have adapted to dung from domestic animals, and thus are more abundant at highly disturbed sites [132], even in arid soils. For example, Maldonado et al. [56] compared two native dung beetle species (*Sulcophanaeus imperator* and *Eucranium arachnoides*) to a eurytopic dung beetle species (*Digitonthophagus gazella*) in a disturbed arid forest in Argentina, and found that the generalist species incorporated a greater amount of organic matter, total nitrogen, and phosphorus into the soil. Furthermore, as Menéndez et al. [133] demonstrated, dung beetles with a tunneling behavior are particularly good promoters of microbial activity and respiration in deeper layers of the soil, which is especially relevant in arid lands, where nutrients above the dry soil surface are not available to plants, and are therefore more likely to be lost through the volatilization processes. These results explain the abundance of *C. balteatus, D. problematicus,* and *O. confusus* in the dry forest area of southern Ecuador at highly disturbed sites, because they are tunnellers.

### 3.7. Redundancy Analysis for Dung Beetle Species, Soil and Environmental Attributes, and Land Uses

Figure 4 illustrates the results of the redundancy analysis corresponding to the occurrence of beetles versus soil and abiotic factors for the different L-UCH classes. Where, 77.3% and 22.5% of the restriction inertia were explained by components 1 (RDA 1) and 2 (RDA 2), respectively. RDA 1 was positively related to the chemical properties of the soil (Figure 4a), such as pH, SOM, SOC, TN, K, and P, and L-UCH, CyAS, GLd, and OCA (Figure 4b), but specifically with the species *C. balteatus*, *D. problematicus,* and *O. confusus* (Figure 2a–c and Figure 4c); while temperature, Bd, and altitude were negatively correlated with these species. These findings are consistent with the data reported by Chamorro et al. [30] and Celi and Dávalos [134], who stated that these species can be used in environmental impact studies as bio indicators, because these species inhabit soils with low fertility. Furthermore, these species are essential to the recycling of nutrients and the return of nutrients into the soil, due to their predominant activity in manure processing, decomposition, and incorporation of manure into the soil [125]. Therefore, they are particularly abundant in soils with low SOM content, such as the L-UCH classes CyAS (SOM: 2.1%) and GLd (SOM: 2.3%; Table 3). This was confirmed by Rangel-Acosta et al. [113], who showed that *C. balteatus*, *D. problematicus*, and *O. confusus* are eurytopic species, which can be found especially in different anthropogenically disturbed environments (Table 4). Apparently, low soil fertility and little plant cover is the main requirement for these species, where they support nutrient cycling through the accelerated decomposition of organic residues and manure [127], which is why these species can be used as indicators in altered habitats.

RDA 2 was positively related to altitude, temperature, and Bd (soil compaction) [49], but negatively with SOM, macronutrients, and pH (Figure 3a). The species related to these abiotic factors were *O. curvicornis* and *Ph. achille*, in particular those captured at CoAS and SDF (Figure 3b). As explained before, altitude is generally related to temperature, but in highly disturbed L-UCH classes, temperature becomes more extreme [78], which might affect the adaptation of *O. curvicornis* and *Ph. achille*. Furthermore, Bd could be another limiting factor for these species, as shown by Brussaard and Slager [125], who observed that food preparation and nesting is more time-consuming for dung beetles in soil with low Bd (see Section 3.6). The RDA explains why fewer eggs and fewer hatchlings were found in less compacted soils compared to dense soils, and why these species were less abundant in all the L-UCH classes studied (Table 4).

## 4. Conclusions

In this study, three factors were correlated to abundance and diversity of dung beetles in a disturbed TDF region in southern Ecuador: (i) the level of disturbance, represented here by different L-UCH classes (CyAS, GLd, OCA, SDF, and CoAS); (ii) abiotic factors (air temperature and altitude), and iii) soil fertility (physical–chemical properties). Due to the altered environmental conditions in all L-UCH classes and the availability of food (dung), most of the species caught were eurytopic. These species inhabit more compact soils (high Bd), which generally occur in highly disturbed anthropogenic sites, due to the use of agricultural machinery (CyAS), as well as high stock density (up to 5 animals per hectare), which leads to overgrazing, especially at pasture sites (GLd). Furthermore, the application of dung and organic fertilizers at these sites improves their food availability. On the other hand, in less disturbed areas (SDF and CoAS) fewer generalist species were found, and native dung beetle species were more numerous (e.g., *Onthophagus curvicornis*). These results suggest that anthropogenic land-use change affects the community structure of dung beetles in TDF ecosystems, favoring eurytopic or generalist dung beetle species. The most abundant dung beetle species in the study area were *C. balteatus, D. problematicus,* and *O. confuses,* which were captured particularly in highly disturbed sites (CyAS, GLd, and OCA). Therefore, the abundance and diversity of dung beetles depend on the level of disturbance, which includes food supply, because manure from introduced domestic animals favors the distribution of eurytopic species. These findings can help decision makers to design policies and technical proposals for environmental restoration in the TDFs of southern Ecuador.

## Figures and Tables

**Figure 1 insects-12-00306-f001:**
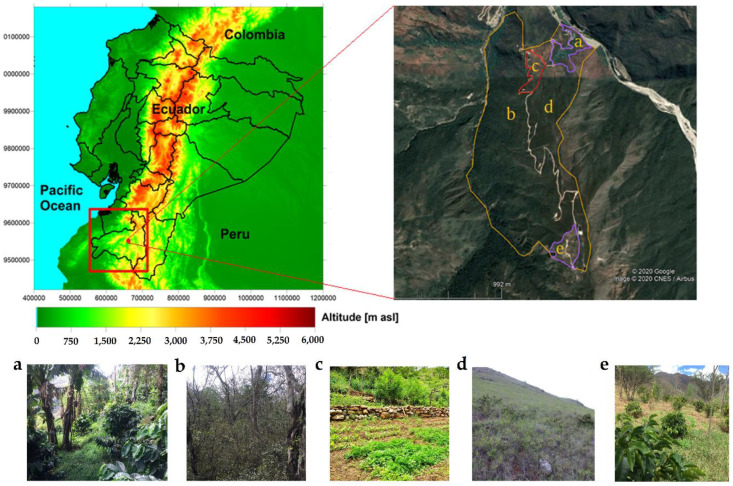
Digital elevation model (DEM) of continental Ecuador (left) and study area (right). Below: different land-use changes (L-UCH) classes analyzed in this study, where (**a**) CoAS = coffee in an old agroforestry system; (**b**) SDF = disturbed seasonally dry forest; (**c**) OCA = organic crop area; (**d**) GLd = grassland; and (**e**) CyAS = coffee in a young agroforestry system.

**Figure 2 insects-12-00306-f002:**
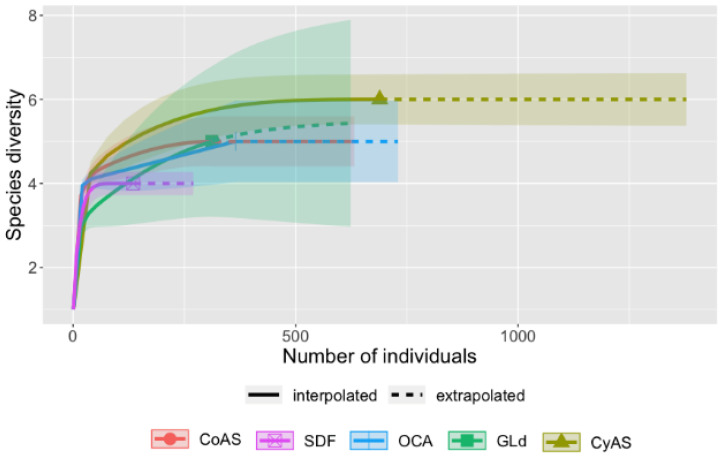
Extrapolation curves of the assemblages of *Scarabaeinae* sampled for five land uses, Ecuador.

**Figure 3 insects-12-00306-f003:**
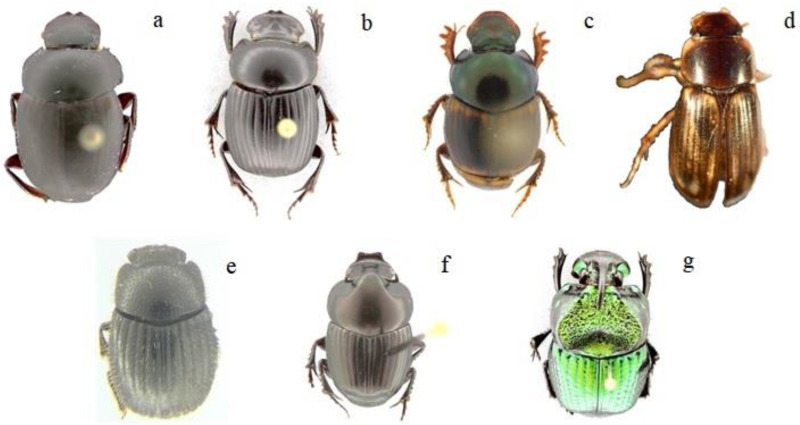
Dung beetle species found in different L-UCH classes in a disturbed tropical dry forest (TDF) ecosystem in southern Ecuador. (**a**). *Canthon balteatus*; (**b**). *Dichotomius problematicus*; (**c**). *Onthophagus confusus*; (**d**). *Aphodius* sp1; (**e**). *Onoreidium ohausi*; (**f**). *Onthophagus curvicornis*; (**g**). *Phanaeus achilles*.

**Figure 4 insects-12-00306-f004:**
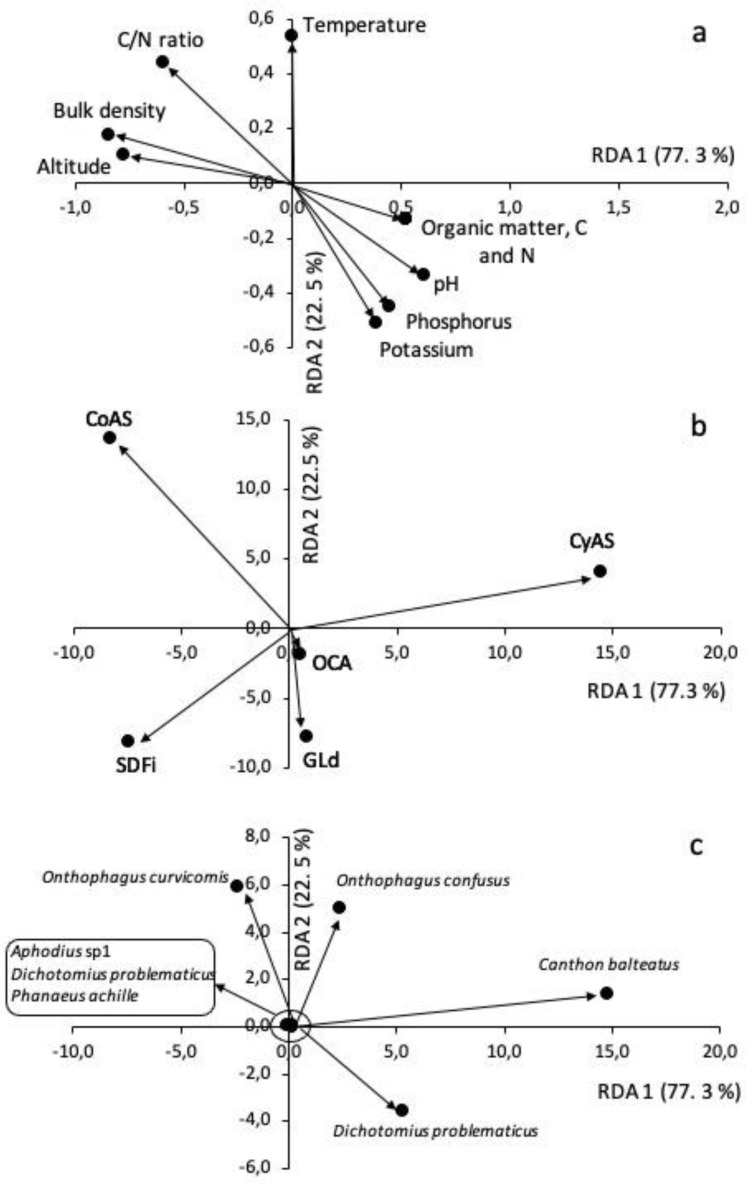
Scores of the soil and environmental variables (**a**), land uses (**b**), and dung beetle species (**c**) of the redundancy analysis (components: RDA 1 and RDA 2). Note the different scales of RDA 1 and RDA 2.

**Table 1 insects-12-00306-t001:** Altitude of the different L-UCH classes and mean air temperature at each site during the whole study period (May 2019–February 2020).

L-UCH	Altitude (m a.s.l.)	Temperature (°C)
Coffee in an old agroforestry system (CoAS)	1404	22.1
Seasonally dry forest disturbed (SDF)	1525	21.6
Organic crop area (OCA)	1558	22.1
Grassland (GLd)	1861	22.4
Coffee in a young agroforestry system (CyAS)	1959	20.6

**Table 2 insects-12-00306-t002:** Average values of physical properties of the soil for the different L-UCH classes, including their standard deviations (15 repetitions for each L-UCH class). The different letters mean significant difference (*p* < 0.05, HSD Tukey).

Land Use Pattern	Bulk Density	Sand	Silt	Clay	Textural Class
	(g cm^−3^)	%	%	%	
CoAS	0.80 ± 0.1 ^a^	69.2 ± 3.1 ^a^	16.0 ± 3.5 ^b^	14.8 ± 6.4 ^a^	Loam-sandy soil
SDF	0.71 ± 0.1 ^a^	77.2 ± 6.1 ^a^	11.3 ± 3.1 ^ab^	11.4 ± 5.3 ^a^	Loam-sandy soil
OCA	0.92 ± 0.1 ^ab^	74.6 ± 2.0 ^a^	12.7 ± 1.2 ^ab^	12.8 ± 1.2 ^a^	Loam-sandy soil
GLd	1.12 ± 0.1 ^b^	75.8 ± 2.0 ^a^	7.3 ± 1.2 ^a^	16.2 ± 0.0 ^a^	Loam-sandy soil
CyAS	1.05 ± 0.1 ^b^	70.5 ± 3.1 ^a^	11.3 ± 4.21 ^ab^	18.2 ± 2.0 ^a^	Loam-sandy soil

**Table 3 insects-12-00306-t003:** Average values of chemical properties of the soil for the different L-UCH classes, including their standard deviations (15 repetitions for each L-UCH class). The different letters mean significant difference (*p* < 0.05, HSD Tukey). SOM: soil organic matter. TN: total nitrogen. SOC: soil organic carbon.

Land Use Pattern	pH	SOM	TN	SOC	C/N Ratio	P	k
		%	%	%		(mg/kg)	(cmol/kg)
CoAS	6.03 ± 0.5 ^b^	6.7± 2,9 ^b^	0.34 ± 0.15 ^b^	3.9 ± 1.73 ^b^	11.5 ± 0.2 ^a^	31.8 ± 39.5 ^ab^	0.66± 0.5 ^ab^
SDF	6.0 ± 0.6 ^b^	4.5 ± 0.9 ^ab^	0.23 ± 0.05 ^ab^	2.6 ± 0.56 ^ab^	11.3 ± 0.2 ^a^	37.6 ± 60.8 ^ab^	0.54 ± 0.4 ^ab^
OCA	6.7 ± 0.4 ^b^	3.7± 1.5 ^ab^	0.18 ± 0.07 ^ab^	2.2 ± 0.89 ^ab^	11.6 ± 0.1 ^a^	87.5 ± 34.6 ^b^	1.32± 0.7 ^b^
GLd	5.2 ± 0.3 ^a^	2.3± 0.7 ^a^	0.11 ± 0.04 ^a^	1.3 ± 0.44 ^a^	11.8 ± 0.8 ^a^	4.6 ± 0.7 ^a^	0.13± 0.1 ^a^
CyAS	5.01 ± 0.2 ^a^	2.1± 0.8 ^a^	0.11 ± 0.04 ^a^	1.2 ± 0.46 ^a^	11.4 ± 0.6 ^a^	4.3 ± 1.3 ^a^	0.18± 0.2 ^a^

**Table 4 insects-12-00306-t004:** Dung beetle species and the average number of individuals trapped in the different L-UCH classes, including the standard deviations (24 pitfall traps in each L-UCH class). The different letters mean significant difference (*p* < 0.05, HSD Tukey).

Species	Land Uses					*p*-Value
	CoAS	SDF	OCA	GLd	CyAS	
*Aphodius* sp1	0.08 ± 0.3 ^a^	0.0 ± 0.0 ^a^	0.04 ± 0.2 ^a^	0.0 ± 0.0 ^a^	0.0 ± 0.0 ^a^	0.248
*Canthon balteatus* (Boheman, 1858)	1.0 ± 0.4 ^b^	0.3 ± 0.6 ^b^	5.5 ± 2.0 ^ab^	5.4 ± 1.9 ^ab^	24.4 ± 7.9 ^a^	0.000
*Dichotomius problematicus* (Luederwaldt, 1923)	1.2 ± 0.6^b^	4.5 ± 1.7 ^ab^	5.7 ± 1.9 ^ab^	6.4 ± 1.8 ^ab^	9.9 ± 4.4 ^a^	0.000
*Onoreidium ohausi* (Arrow, 1931)	0.0 ± 0.0 ^b^	0.0 ± 0.0 ^b^	0.0 ± 0.0 ^b^	0.0 ± 0.0 ^b^	0.2 ± 0.0 ^a^	0.002
*Onthophagus confusus* Boucomont, 1932	4.7 ± 2.8 ^a^	0.3 ± 0.5 ^b^	2.3 ± 1.3 ^ab^	1.1 ± 0.4 ^ab^	5.2 ± 2.3 ^a^	0.000
*Onthophagus curvicornis* Latreille, 1811	6.1 ± 2.0 ^b^	0.5 ± 0.7 ^a^	1.7 ± 2.4 ^ab^	0.0 ± 0.2 ^a^	1.6 ± 0.8 ^ab^	0.000
*Phanaeus achilles* Boheman, 1858	0.0 ± 0.0 ^a^	0.0 ± 0.0 ^a^	0.0 ± 0.0 ^a^	0.1 ± 0.3 ^a^	0.2 ± 0.2 ^a^	0.056
Total number of species found	5	4	5	4	6	---
Shannon_H´	1.1 ± 0.2 ^b^	1.1 ± 0.1 ^b^	1.0 ± 0.2 ^b^	0.9 ± 0.1 ^b^	0.4 ± 0.4 ^a^	0.000
Simpson	0.4 ± 0.1 ^ab^	0.8 ± 0.3 ^b^	0.3 ± 0.0 ^a^	0.4 ± 0.0 ^ab^	0,4 ± 0.4 ^ab^	0.000

**Table 5 insects-12-00306-t005:** Main vertebrates (native and domestic) inhabiting the different L-UCH classes and use of manure types. Mean values (X) close to 1 indicate frequent occurrence of these types of animal, while values around 0 indicate absence or less frequent occurrence of these types of animal.

Type of Vertebrate/Uses of Manure	CoAS		SDF		OCA		GLd		CyAS	
	X	Ca (ind./ha)	X	Ca (ind./ha)	X	Ca (ind./ha)	X	Ca (ind./ha)	X	Ca (ind./ha)
Native vertebrates										
Anteater	0.3	Nd	1.0	Nd	0.0	Nd	0.0	Nd	0.3	Nd
Deer	0.4	Nd	1.0	Nd	0.1	Nd	0.0	Nd	0.1	Nd
Skunk	0.0	Nd	1.0	Nd	0.1	Nd	0.1	Nd	0.3	Nd
Opossum	0.1	Nd	1.0	Nd	0.0	Nd	0.0	Nd	0.0	Nd
Squirrels	0.4	Nd	0.4	Nd	0.0	Nd	0.0	Nd	0.0	Nd
Exotic vertebrates										
Goats	0.4	5 ind/1 hectare/1 month	1.0	5 ind./hectare/1 months	0.0		0.4	3 ind/hectare/1 month	0.0	
Cows	0.0		0.3	2 ind./hectare/1 months	0.0		1.0	1 ind/hectare/1 month	0.8	1 ind/hectare/1 month
Donkeys	0.0		0.1	2 ind./hectare/1 months	0.4	4 ind/1 hectare/1 month	0.8	5 ind/1 hectare/1 month	0.0	
Horses	0.0		0.3	2 ind./hectare/1 months	0.3	4 ind/1 hectare/1 month	0.9	5 ind/1 hectare/1 month	0.6	1 ind/hectare/1 month
Uses of manure	CoAS		SDF		OCA		GLd		CyAS	
	X		X		X		X		X	
Do you apply dung from domestic animals directly to the soil?	0.8		0.0		0.9		0.9		0.8	
Do you use organic fertilizers?	0.5		0.0		0.9		0.9		0.9	
Do you use human feces?	0.5		0.0		0.5		0.4		0.8	

X = Average of farmers’ responses; Ca = Approximate animal load; Nd = No data.

**Table 6 insects-12-00306-t006:** Effects of abiotic, edaphic properties, and type factors of L-UCH on the total abundance of Scarabaeinae species.

Response Variable	Explanatory Variable	Error Standard	Z-Value	*p*-Value
Total abundance	Abiotic factors			
	Altitude	0.000	20.584	<0.001
	Temperature	0.027	21.320	<0.001
	Edaphic properties			
	Bd	0.146	15.870	<0.001
	pH	0.037	−15.880	<0.001
	SOM	0.015	−14.480	<0.001
	N	0.293	−13.980	<0.001
	P	0.001	−10.480	<0.001
	K	0.054	−10.850	<0.001
	C	0.026	−14.430	<0.001
	C/N ratio	0.140	−1.343	0.179
	Land Uses			
	CoAS	0.055	72.466	<0.001
	CyAS	0.064	16.323	<0.001
	GLd	0.078	0.078	0.938
	OCA	0.075	2.488	0.013
	SDFi	0.097	−7.493	<0.001

## Data Availability

The research data were deposited in the Southern Ecuador Insect Collection [CISEC-MUTPL].

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
