# Peer review of "Effects of Land-Use Change on the Community Structure of the Dung Beetle (Scarabaeinae) in an Altered Ecosystem in Southern Ecuador"

_insects, 2021, doi:10.3390/insects12040306_

Round 1
Reviewer 1 Report
Assessment of the paper entitled “Effects of land-use change on the community structure of the dung beetle (Scarabaeinae) in an altered ecosystem in southern Ecuador” for Insects (insects-1042958)
Main comments
In this paper, the authors aimed to evaluate the effects of land-use changes on dung beetle communities in a semiarid ecosystem from Ecuador. After reading the paper, and given the current knowledge on the issue (e.g., Barretto et al., 2020; Hernández, 2005, 2007; Liberal et al., 2011; Medina & Lopes, 2014; Neves et al., 2010; Vieira et al., 2017), I thought the authors need to improve the statistical approach and theoretical background of their study. I suggest the authors addressing the most recent literature on dung beetle response to land-use changes, mainly in semiarid ecosystems (some examples cited above). Besides, the study needs a more hypothesis-related approach to improve readership appealing. Finally, I strongly suggest the authors improving the information on the methods’ description. The reader is unable to understand many aspects of the sampling design and statistical analyses used. Regarding the latter, the authors could improve their analytical approach by using generalized linear models, which are adequate to model all kinds of data without using severe data transformation. In addition, the use of redundancy analysis (RDA) or PERMANOVA could be better approached (although different) to relate community composition with abiotic data than principal component analysis (Anderson & Walsh, 2013; Borcard et al., 2018).
References
Anderson, M.J. & Walsh, D.C.I. (2013) PERMANOVA, ANOSIM, and the Mantel test in the face of heterogeneous dispersions: What null hypothesis are you testing? Ecological Monographs, 83, 557-574.
Barretto, J., Salomão, R.P., & Iannuzzi, L. (2020) Diversity of dung beetles in three vegetation physiognomies of the Caatinga dry forest. International Journal of Tropical Insect Science, 40, 385-392.
Borcard, D., Gillet, F., & Legendre, P. (2018) Numerical ecology with R, 2nd edn. Springer, New York.
Hernández, M.I.M. (2005). Besouros Scarabaeidae (Coleoptera) da área do Curimataú, Paraíba. In Análise das variações da biodiversidade do bioma Caatinga: Suporte a estratégias regionais de conservação (ed. by F.S. Araújo, M.J.N. Rodal & M.R.V. Barbosa), pp. 369-380. Ministério do Meio Ambiente, Brasília.
Hernández, M.I.M. (2007) Besouros escarabeíneos (Coleoptera: Scarabaeidae) da Caatinga Paraibana, Brasil. Oecologia Brasiliensis, 11, 356-364.
Liberal, C.N., Farias, Â.M.I., Meiado, M.V., Filgueiras, B.K.C., & Iannuzzi, L. (2011) How habitat change and rainfall affect dung beetle diversity in Caatinga, a Brazilian semi-arid ecosystem. Journal of Insect Science, 11, 114.
Medina, A.M. & Lopes, P.P. (2014) Resource utilization and temporal segregation of Scarabaeinae (Coleoptera, Scarabaeidae) community in a Caatinga fragment. Neotropical Entomology, 43, 127-133.
Neves, F.d.S., Oliveira, V.H.F., Espírito-Santo, M.M., Vaz-de-Mello, F.Z., Louzada, J., Sanchez-Azofeifa, A., & Fernandes, G.W. (2010) Successional and seasonal changes in a community of dung beetles (Coleoptera: Scarabaeinae) in a Brazilian tropical dry forest. Natureza & Conservação, 8, 160-164.
Vieira, L., Silva, F.A.B., & Louzada, J. (2017) Dung beetles in a Caatinga Natural Reserve: a threatened Brazilian dry-forest with high biological value. Iheringia Serie Zoologia, 107, e2017045.
Author Response
Reviewer 1
Open Review
(x) I would not like to sign my review report
( ) I would like to sign my review report
English language and style
(x) Extensive editing of English language and style required
( ) Moderate English changes required
( ) English language and style are fine/minor spell check required
( ) I don't feel qualified to judge about the English language and style
The English language was revised by a native speaker.
Yes |
Can be improved |
Must be improved |
Not applicable |
|
Does the introduction provide sufficient background and include all relevant references? |
( ) |
( ) |
(x) |
( ) |
Is the research design appropriate? |
( ) |
( ) |
(x) |
( ) |
Are the methods adequately described? |
( ) |
( ) |
(x) |
( ) |
Are the results clearly presented? |
( ) |
( ) |
(x) |
( ) |
Are the conclusions supported by the results? |
( ) |
( ) |
(x) |
( ) |
Comments and Suggestions for Authors
Assessment of the paper entitled “Effects of land-use change on the community structure of the dung beetle (Scarabaeinae) in an altered ecosystem in southern Ecuador” for Insects (insects-1042958)
All sections of the manuscript were revised and improved.
Main comments
In this paper, the authors aimed to evaluate the effects of land-use changes on dung beetle communities in a semiarid ecosystem from Ecuador. After reading the paper, and given the current knowledge on the issue (e.g., Barretto et al., 2020; Hernández, 2005, 2007; Liberal et al., 2011; Medina & Lopes, 2014; Neves et al., 2010; Vieira et al., 2017), I suggest the authors addressing the most recent literature on dung beetle response to land-use changes, mainly in semiarid ecosystems (some examples cited above).
Additional information about recent investigations were included in the introduction, as well as in the results and discussion section.
I thought the authors need to improve the statistical approach and theoretical background of their study.
The statistical analysis was changed (previously: principal component analysis [PCA]) as suggested by the reviewer, and a general linear model (GLM) and a redundancy analysis (RDA) applied. The method section was reworked and each single step clarified. Also, the statistical approaches were explained in more detail (see also last comment).
Besides, the study needs a more hypothesis-related approach to improve readership appealing.
The hypothesis of the study was clarified in the introduction section and connected to the used approaches. Furthermore, the conclusion section was reworked and the focus set on the demonstration of the hypothesis.
Finally, I strongly suggest the authors improving the information on the methods’ description. The reader is unable to understand many aspects of the sampling design and statistical analyses used. Regarding the latter, the authors could improve their analytical approach by using generalized linear models, which are adequate to model all kinds of data without using severe data transformation. In addition, the use of redundancy analysis (RDA) or PERMANOVA could be better approached (although different) to relate community composition with abiotic data than principal component analysis (Anderson & Walsh, 2013; Borcard et al., 2018).
The method section was reworked and the single steps clarified. Also, the statistical approaches were explained in more detail. As suggested by the reviewer, the statistical analysis was changed (previously: principal component analysis [PCA]), and a general linear model (GLM) and a redundancy analysis (RDA) applied.

Reviewer 2 Report
- Authors must change the abstract. Now it does not reflect the results, but only provides General information. We need to specify the results.
- You need to add a few more links to the Introduction. The reference is required to reduce the species diversity of plants due to anthropogenic activities (Juiling S., Leon S.K., Jumian J., Tsen S., Lee Y.L., Khoo E., Sugau J.B., Nilus R., Pereira J.T., Damit A., Tanggaraju S., O'Byrne P., Sumail S., Mujih H., Maycock C.R. 2020. Conservation assessment and spatial distribution of endemic orchids in Sabah, Borneo. Nature Conservation Research. Vol. 5(Suppl.1). P. 136–144. https://dx.doi.org/10.24189/ncr.2020.053), changes in the number of species due to poaching (de Lima N.S., Napiwoski S.J., Oliveira M.A. 2020. Human-wildlife conflict in the Southwestern Amazon: poaching and its motivations. Nature Conservation Research. Vol. 5(1). P. 109–114. https://dx.doi.org/10.24189/ncr.2020.006), the impact of highways on animals (Mohd-Azlan J., Lok L., Maiwald M.J., Fazlin S., Shen T.D., Kaicheen S.S., Dagang P. 2020. The distribution of medium to large mammals in Samunsam Wildlife Sanctuary, Sarawak in relation to the newly constructed Pan-Borneo Highway. Nature Conservation Research. Vol. 5(4). P. 43–54. https://dx.doi.org/10.24189/ncr.2020.055), and reducing species diversity due to human activities (Chifundera K.Z. 2019. Using diversity indices for identifying the priority sites for herpetofauna conservation in the Democratic Republic of the Congo. Nature Conservation Research. Vol. 4(3). P. 13–33. https://dx.doi.org/10.24189/ncr.2019.035 ).
- In the Results and discussion section, there are links that are not made according to the rules of the journal (for example, Fries et al., 2009). It is necessary to give more information about the influence of factors on beetles. There is no section 3.4.
- Сonclusion needs to be redone. It should describe how the factors studied affect each species. It is also necessary to briefly describe the variability of physical factors, based on sections 3.1-3.3.
Author Response
Reviewer 2
Open Review
(x) I would not like to sign my review report
( ) I would like to sign my review report
English language and style
( ) Extensive editing of English language and style required
( ) Moderate English changes required
( ) English language and style are fine/minor spell check required
(x) I don't feel qualified to judge about the English language and style
Yes |
Can be improved |
Must be improved |
Not applicable |
|
Does the introduction provide sufficient background and include all relevant references? |
( ) |
(x) |
( ) |
( ) |
Is the research design appropriate? |
(x) |
( ) |
( ) |
( ) |
Are the methods adequately described? |
(x) |
( ) |
( ) |
( ) |
Are the results clearly presented? |
(x) |
( ) |
( ) |
( ) |
Are the conclusions supported by the results? |
( ) |
(x) |
( ) |
( ) |
Comments and Suggestions for Authors
- Authors must change the abstract. Now it does not reflect the results, but only provides General information. We need to specify the results.
The abstract was reworked and the results specified.
- You need to add a few more links to the Introduction. The reference is required to reduce the species diversity of plants due to anthropogenic activities (Juiling S., Leon S.K., Jumian J., Tsen S., Lee Y.L., Khoo E., Sugau J.B., Nilus R., Pereira J.T., Damit A., Tanggaraju S., O'Byrne P., Sumail S., Mujih H., Maycock C.R. 2020. Conservation assessment and spatial distribution of endemic orchids in Sabah, Borneo. Nature Conservation Research. Vol. 5(Suppl.1). P. 136–144. https://dx.doi.org/10.24189/ncr.2020.053), changes in the number of species due to poaching (de Lima N.S., Napiwoski S.J., Oliveira M.A. 2020. Human-wildlife conflict in the Southwestern Amazon: poaching and its motivations. Nature Conservation Research. Vol. 5(1). P. 109–114. https://dx.doi.org/10.24189/ncr.2020.006), the impact of highways on animals (Mohd-Azlan J., Lok L., Maiwald M.J., Fazlin S., Shen T.D., Kaicheen S.S., Dagang P. 2020. The distribution of medium to large mammals in Samunsam Wildlife Sanctuary, Sarawak in relation to the newly constructed Pan-Borneo Highway. Nature Conservation Research. Vol. 5(4). P. 43–54. https://dx.doi.org/10.24189/ncr.2020.055), and reducing species diversity due to human activities (Chifundera K.Z. 2019. Using diversity indices for identifying the priority sites for herpetofauna conservation in the Democratic Republic of the Congo. Nature Conservation Research. Vol. 4(3). P. 13–33. https://dx.doi.org/10.24189/ncr.2019.035 ).
The introduction section was revised and extended. Also, the suggested references were included.
- In the Results and discussion section, there are links that are not made according to the rules of the journal (for example, Fries et al., 2009). It is necessary to give more information about the influence of factors on beetles. There is no section 3.4.
The result and discussion section was revised and the findings clarified. Also an image of the dung beetle species found in the study area was added. The formal errors with respect to the journal rules were eliminated.
- Сonclusion needs to be redone. It should describe how the factors studied affect each species. It is also necessary to briefly describe the variability of physical factors, based on sections 3.1-3.3.
The conclusion section was reworked and adjusted to demonstrate the hypothesis. Also a brief description of the variability of the physical factors were included.

Reviewer 3 Report
This is an extremely thorough experiment and ms. that has been well planned and carefully interpreted. Is is worthy of publication with minor editorial changes.
I have only a few editorial comments to suggest:
intervened is a literal interpretation of the Spanish intervenido and is better translated as disturbed or disturbance
171- pit fall traps
259 - delet the use of very
310 - natural should be replaced with a more appropriate word such as intact
311 - delet natural
419 - delete which determine and replace it with that influence the abundance
Author Response
Reviewer 3:
Open Review
( ) I would not like to sign my review report
(x) I would like to sign my review report
English language and style
( ) Extensive editing of English language and style required
( ) Moderate English changes required
(x) English language and style are fine/minor spell check required
( ) I don't feel qualified to judge about the English language and style
The English language was revised by a native speaker.
Yes |
Can be improved |
Must be improved |
Not applicable |
|
Does the introduction provide sufficient background and include all relevant references? |
(x) |
( ) |
( ) |
( ) |
Is the research design appropriate? |
(x) |
( ) |
( ) |
( ) |
Are the methods adequately described? |
(x) |
( ) |
( ) |
( ) |
Are the results clearly presented? |
(x) |
( ) |
( ) |
( ) |
Are the conclusions supported by the results? |
(x) |
( ) |
( ) |
( ) |
Comments and Suggestions for Authors
This is an extremely thorough experiment and ms. that has been well planned and carefully interpreted. Is is worthy of publication with minor editorial changes.
Thank you!
I have only a few editorial comments to suggest:
intervened is a literal interpretation of the Spanish intervenido and is better translated as disturbed or disturbance
It was adjusted throughout the text.
171- pit fall traps
259 - delet the use of very
310 - natural should be replaced with a more appropriate word such as intact
311 - delet natural
419 - delete which determine and replace it with that influence the abundance
All sections of the manuscript were revised and improved. The suggested changes were realized.
